# An Updated Review of *Ornithodoros* Ticks as Reservoirs of African Swine Fever in Sub-Saharan Africa and Madagascar

**DOI:** 10.3390/pathogens12030469

**Published:** 2023-03-16

**Authors:** Ferran Jori, Armanda Bastos, Fernando Boinas, Juanita Van Heerden, Livio Heath, Hélène Jourdan-Pineau, Beatriz Martinez-Lopez, Rémi Pereira de Oliveira, Thomas Pollet, Carlos Quembo, Keaton Rea, Edgar Simulundu, Florian Taraveau, Mary-Louise Penrith

**Affiliations:** 1UMR ASTRE (Animal, Health, Territories, Risks and Ecosystems), CIRAD Campus International de Baillarguet, 34398 Montpellier, France; 2Department of Zoology and Entomology, Faculty of Natural and Agricultural Sciences, University of Pretoria, Pretoria 0028, South Africa; 3CIISA—Centre for Interdisciplinary Research in Animal Health, Faculty of Veterinary Medicine, University of Lisbon, 1300-477 Lisbon, Portugal; 4Associate Laboratory for Animal and Veterinary Science (AL4AnimalS), 1300-477 Lisbon, Portugal; 5Transboundary Animal Diseases, Onderstepoort Veterinary Research, Agricultural Research Council, Pretoria 0110, South Africa; 6Center for Animal Disease Modeling and Surveillance (CADMS), School of Veterinary Medicine, University of California, Davis, CA 95616, USA; 7Central Region Office—Regional Veterinary Laboratory, Agricultural Research Institute of Mozambique, Chimoio EN6.CP42, Mozambique; 8Macha Research Trust, Choma 20100, Zambia; 9Department of Disease Control, School of Veterinary Medicine, University of Zambia, Lusaka 10101, Zambia; 10Department of Veterinary Tropical Diseases, Faculty of Veterinary Science, University of Pretoria, Onderstepoort, Pretoria 0110, South Africa

**Keywords:** sylvatic cycle, warthog, vector competency, distribution, *p72* genotypes, East Africa, Southern Africa, endemic, argasid, ecology

## Abstract

This updated review provides an overview of the available information on *Ornithodoros* ticks as reservoirs and biological vectors of the ASF virus in Africa and Indian Ocean islands in order to update the current knowledge in this field, inclusive of an overview of available methods to investigate the presence of ticks in the natural environment and in domestic pig premises. In addition, it highlights the major areas of research that require attention in order to guide future investigations and fill knowledge gaps. The available information suggests that current knowledge is clearly insufficient to develop risk-based control and prevention strategies, which should be based on a sound understanding of genotype distribution and the potential for spillover from the source population. Studies on tick biology in the natural and domestic cycle, including genetics and systematics, represent another important knowledge gap. Considering the rapidly changing dynamics affecting the African continent (demographic growth, agricultural expansion, habitat transformation), anthropogenic factors influencing tick population distribution and ASF virus (ASFV) evolution in Africa are anticipated and have been recorded in southern Africa. This dynamic context, together with the current global trends of ASFV dissemination, highlights the need to prioritize further investigation on the acarological aspects linked with ASF ecology and evolution.

## 1. Introduction

African swine fever (ASF) is undoubtedly the most serious disease currently affecting the global pig production sector. The ASF virus (ASFV), which exclusively affects members of the pig/Suidae family, is able to persist for several weeks in a protein-rich environment protected from sunlight and desiccation [1]. African wild pigs such as the common warthog (*Phacochoerus africanus*) and bushpigs (*Potamochoerus larvatus*) have been found to be resistant to the disease. However, domestic pigs and Eurasian wild boars (*Sus scrofa*) are susceptible to infection, and the virus induces very high mortality rates, usually within a matter of days after first contact of *Sus* hosts with the virus [1,2]. In addition, there is no wide-scale vaccine available to immunize the pig population, and the most efficient way to prevent transmission and maintenance of the disease is prevention through risk mitigation measures based on a better understanding of disease epidemiology and ecology [1,2].

ASFV genotype II, introduced from East Africa into the Caucasus in 2007, has expanded its range both outside and within sub-Saharan Africa [1]. The expansion of the global footprint of this disease has triggered scientific attention and generated a considerable amount of information on the eco-epidemiology of the disease in newly-infected regions, particularly with reference to the ecological role of wild boars in the EU [2]. However, this virus, which was first described in Africa a century ago [3], has existed in an ancient sylvatic cycle involving warthogs and the soft ticks that infest their burrows for millennia. It was only when susceptible domestic suids were first introduced into the African continent that the disease, the role of wild suids, and two additional epidemiological cycles were recognized: one exclusively involving domestic pigs and another involving virus cycling between domestic pigs and the soft tick inhabiting their pens [4]. 

The long-term co-evolution between the virus, warthogs, and soft ticks confirms a very well-established involvement of these vectors in the natural environment in Africa. This natural cycle involving soft ticks represents a permanent, albeit sporadic, source of ASFV for the domestic pig value chain. In this context, the potential eradication of ASF from the parts of Africa where the sylvatic cycle is present, even should a vaccine be available, is an unrealistic goal, and the only viable strategy is to prevent spillover from the sylvatic cycle to domestic hosts and to control the virus once introduced to domestic pig populations.

*Ornithodoros* soft ticks, also known as tampans, are the only known arthropod biological vectors of ASFV, and they are morphologically and biologically different from hard ticks (Figure 1). 

The association of eyeless *Ornithodoros moubata s.l.* ticks with warthogs is well described [5,6]. Further eyeless species were described subsequently, but the taxonomy is complicated, and for convenience, these ticks are widely referred to as the *Ornithodoros moubata* complex [4]. These ticks are cryptic and photophobic, with short feeding times of generally less than an hour, spending most of their lives in burrows and crevices in the vertebrate hosts’ resting places. Apart from transmitting the virus to suid species when feeding on their blood, they are able to transmit the ASF virus transovarially, trans-stadially, and sexually [5,7,8], thus maintaining the virus in the absence of an infectious suid vertebrate host bloodmeal. Replication in midgut epithelium, circulating hemocytes, and salivary and coxal glands have been described in detailed laboratory studies on *O. moubata s.l.* ticks derived from both field collections and a laboratory colony [9,10,11,12].

These laboratory studies reported transmission of ASFV for at least 15 months after a blood meal in ticks from warthog burrows and suggested that longer transmission periods should be investigated [9,10,11,12]. A study of *O. erraticus* indicated that virus transmission was possible after five years without a blood meal and reported longevity of ticks for 15–20 years [13]. Although most of the *Ornithodoros* species studied experimentally have been able to maintain ASFV and transmit it to pigs, transovarial transmission has not been demonstrated in all species evaluated to date [14,15,16]. Vector competence remains a topic for investigation [17,18,19] and is discussed in more detail in Section 5.

In the last decades, the domestic pig population within the African continent has grown exponentially, leading to the domestic pig cycle playing a major role in disease maintenance and dissemination in Africa. A domestic pig-tick cycle, first discovered in Spain in the 1960s [20], has occasionally been reported in some African countries and possibly in Madagascar [21]. However, the extent of its occurrence on the African continent and its potential role as a source of ASF outbreaks in domestic pigs has received limited attention. Similarly, since the 1980s, studies on the distribution of the sylvatic cycle of ASF in Africa have been scarce. Nevertheless, in recent years, the level of knowledge regarding the viral diversity hosted within the warthog-tick cycle has increased considerably in some countries [22,23,24]. As a result, many questions remain to be answered to have a full understanding of the role of soft ticks in the maintenance and evolution of new ASFV strains.

The goal of this study is to review the available information on the *Ornithodoros* ticks as reservoirs and biological vectors of ASF on the African continent and the Indian Ocean island of Madagascar in order to update the current knowledge in this field. In addition, the major areas of research that require attention in order to guide future investigations and fill knowledge gaps are highlighted.

## 2. Geographical Distribution of *Ornithodoros* Ticks in Natural Areas from Sub-Saharan Africa

Since the first description of ASF more than a century ago [3], it was recognized that the ecology of the disease had a strong link with warthogs, and the involvement of one or more arthropod vectors such as fleas or lice was suspected due to the absence of direct transmission from warthogs to pigs. Subsequent investigations during the last half of the 20th century confirmed the presence of sylvatic cycles in Eastern and Southern Africa, where ASFV is maintained in soft tick populations of the genus *Ornithodoros* that infest the burrows used by common warthogs. Studies investigating the natural reservoirs of the disease in Africa and spanning a 50-year period (1969–2019) confirmed ASFV presence in *Ornithodoros* ticks from a number of East and Southern African countries (Table 1 and Table 2).

### 2.1. Southern African

Within the southern African region, South Africa is undoubtedly the country where the sylvatic cycle has been studied in greater detail. The first investigations dating from 1980–1985, produced a considerable amount of information on the warthog-tick cycle [6,46,52]. Based on frequent outbreaks of ASF in domestic pigs from 1935 to 1936 as well as the presence of warthogs and the fact that the majority of the domestic pigs in the area were traditionally reared under partially to totally free-ranging conditions favorable for ASF transmission, an ASF controlled area (Figure 2) was proclaimed in the north-eastern parts of South Africa in 1935, inclusive of large parts of the Limpopo and Mpumalanga provinces that border Botswana, Zimbabwe, and Mozambique [53]. In 1936, warthogs and bushpigs were declared to be a threat to agriculture, and farmers were encouraged to destroy them [54]. However, the manner in which the virus was transmitted from warthogs to domestic pigs remained a mystery [29], as feeding warthog offal to domestic pigs did not result in efficient virus transmission [3,5,26]. This was finally resolved after the discovery in Spain that a species of *Ornithodoros* that inhabited pig shelters was a competent vector for the virus [20]. Following confirmation of ASFV in *Ornithodoros* ticks from warthog burrows in East Africa [33], investigations in South Africa confirmed infection in ticks from warthog burrows in the ASF controlled area of South Africa, as well as high seropositivity in warthogs in the same area [6,46]. The ASF controlled area was expanded to include the northern tip of KwaZulu-Natal (KZN) province after a low number of seropositive warthogs (4%) and infected *Ornithodoros* ticks (0.06%) were collected from burrows in the Mkuze Game Reserve [46]. The continued inclusion of this region in the ASF controlled area was brought into question by a survey of soft ticks from Mkuze conducted three decades later, which found no evidence of the virus in ticks despite an increase in warthog burrow infestation rates [31]. Similarly, tick surveys conducted in landlocked Swaziland (Eswatini), which is bordered by northern KZN in the south and Mozambique and Kruger National Park to the east and north, respectively, failed to detect ASFV [36]. In addition, soft ticks were not detected in 30 warthog burrows distributed across the Ndumo Game Reserve in northern KZN, and no outbreaks of ASF have been reported in domestic pigs in that area [30]. These results underscore prior findings that the co-occurrence of both warthogs and ticks does not imply virus presence and that some areas inhabited by warthogs are free of ticks [55].

In 2016, Magadla and co-authors evaluated the validity of the ASF controlled area by sampling ticks 20 km north and south of the redline of the ASF controlled area in South Africa between 2008 and 2012. Their results indicated little change in the original demarcated limits of the endemically infected area [53]. Although *Ornithodoros* ticks were found in warthog burrows on wildlife farms within 20 km beyond the boundary of the controlled area in North West Province, Gauteng, Limpopo, and Mpumalanga, ASFV was only detected in a single pool of ticks collected in Mpumalanga south of the redline [53]. These results are consistent with the sporadic nature of ASF outbreaks within the ASF controlled area up until 2011 [22], which primarily involved genotypes associated with the sylvatic cycle and which were readily controlled. However, in 2012, ASF was detected in pigs originating from a farm south of the ASF controlled area, and there have been numerous subsequent outbreaks (2016–2023) outside of the controlled area, involving genotypes rarely associated with ASF outbreaks in South Africa, viz. genotypes I and II, up until that point [56,57]. This dramatic shift in the epidemiology of ASF outbreaks in domestic pigs and recent evidence of ASF infection in warthogs in the Northern Cape Province [58] suggests that ASFV spill-over to the warthog population, with the probable establishment of the warthog-tick sylvatic cycle, has occurred in areas distant from the ASF controlled area. Molecular screening of ticks from warthog burrows at a large number of localities in South Africa revealed the presence of genotype Ia and Ic variants in ticks from warthog burrows, as well as a genotype III strain in warthog burrow ticks in the Free State province [59]. A possible explanation for this extralimital establishment of sylvatic cycle ASFV is that seronegative warthogs translocated to areas outside the controlled area [37] were infested with *Ornithodoros* nymphs, as has been reported previously [60,61,62]. These co-translocated soft ticks, once established at new sites, can act as reservoirs for any ASF viruses newly introduced to areas outside of the ASF controlled area.

A similar resurgence in interest in the sylvatic cycle has been observed in other countries within southern Africa. In Zambia, ASF was first reported in 1912 in Eastern Province, within an area bordering Malawi and Mozambique that is considered endemic for ASF [23,34]. The sylvatic cycle in Zambia was only investigated at the end of the last century in four wildlife areas across the country [34,63]. *Ornithodoros* ticks were collected from warthog burrows at the four sampling sites (South Luangwa NP, Kafue NP, Sumbu NP, and Livingstone NP), and genomic analysis was carried out using restriction enzyme site mapping [63]. More recently, ASFV-infected ticks were reported in Mosi-oa-Tunya National Park in the Southern Province [23,50]. Whilst comprehensive sequence analysis conducted on genetic data of ASFVs viruses collected in domestic pigs between 1989 and 2015 revealed that genotypes I, II, VIII, and XIV have been involved in causing ASF outbreaks in domestic pigs in Zambia, genotypes XI, XII, and XIII have thus far only been detected in ticks in the country.

The presence of the warthog-tick ASF cycle in Mozambique was long suspected, as repeated outbreaks were reported in the Northern provinces and Beira in the 1960s [44]. The natural warthog-tick cycle was finally confirmed in 2007 at the interface area of the Gorongosa National Park, by different methods, including the analysis of historical data, serological sampling of pigs and warthogs [28], tick sampling in warthog burrows and pigsties, followed by viral analysis. Ticks were shown to be infected with genotype II, genotype V and the newly identified genotype XXIV [29]. Serological evidence and the presence of ASF-infected ticks in pigsties suggested the possibility of the occurrence of a domestic pig-tick cycle as well [28,29]. Since then, very few investigations have been conducted to confirm the presence of a sylvatic cycle in other Mozambican areas. Currently, other conservation areas in the Macossa District Central Mozambique (Coutada 9), Maputo National Park (MNP), and Limpopo National Park (LNP) are being investigated. Warthogs are no longer present in the shared ASF-endemic area of northern Mozambique and Malawi [64]. 

Zimbabwe has reported relatively few outbreaks of ASF, with the earliest report being in 1961, caused by a genotype VIII ASFV [48]. In the early 90s, some outbreaks were investigated and related to genotype I and XVII [65], and more recently, in 2015, genotype II ASFV emerged in domestic pigs in Mashonaland Central Province in northern Zimbabwe [66]. As is the case for other southern African countries, contact with warthogs has been suspected of sparking ASF outbreaks in Zimbabwe. Furthermore, Thomson (1985) reported that *Ornithodoros* sp. ticks were found in warthog burrows at two out of four localities sampled in Zimbabwe (Buffalo Range and Sebungwe), where more than 90 percent of warthogs sampled had antibodies to ASF [6]; the ticks were not sampled for ASFV. ASFV-infected ticks sampled from Victoria Falls and Chiredzi, in the north-east and south-west of Zimbabwe [15] and from Hwange National Park, also in the north-west of the country, have been reported [51]. Although genotype I was confirmed in ticks from the northern Victoria Falls and Hwange region, information is still limited regarding the involvement of the sylvatic cycle in ASF outbreaks in this country, which is not surprising due to the small pig population and the fact that the small commercial pig industry is highly compartmentalized. The 2015 outbreaks were attributed to trade in infected pigs and pork [66], as the sylvatic cycle is not present in the affected area (Dr. Douglas Bruce, personal communication, 2015). 

Eswatini (formerly Swaziland) has never reported an outbreak of ASF, and although sylvatic cycle hosts are present, the virus was not detected in soft ticks sampled from warthog burrows in four nature reserves located in the north-eastern part of the country [36]. Lesotho is situated within the borders of South Africa in an area far south of the ASF controlled area and is mostly bordered by mountains. Therefore, the likelihood of finding a sylvatic cycle in this country is very small.

In Botswana, serologically positive warthogs were reported in the northern region [35] and have been identified in the southern part of the country (Jori, unpublished data). However, intensive studies are lacking, probably due to the fact that Botswana has a very small pig population, and only sporadic outbreaks in domestic pigs have been reported, involving genotypes III and VII in 1999 and 1987, respectively [47,65,67].

In Namibia, a large proportion of outbreaks of ASF in pigs are suspected to be related to direct or indirect contacts with warthogs and from transmission originating from the sylvatic cycle [68,69,70,71], as has been described in other Southern African countries. Initial typing efforts which confirmed genotype I in warthogs [47] and in domestic pig outbreaks in Windhoek in 1989 [65] appeared to support this. However, subsequent typing of the 2018 domestic pig outbreak strains viruses and those present in warthogs from the same region, 15 months after the outbreak, revealed that whilst genotype I viruses were present in both suid species, the strains detected in domestic pigs were distinct from those in wild suids [48].

In Malawi, *Ornithodoros* ticks sampled from a culvert used by warthogs were negative for ASFV, and warthogs were absent from the ASF-endemic area. However, a virus was isolated from a warthog in 1960 [47], and ticks sampled from domestic pig pens and from houses were positive for ASFV [42,43]. The historical presence of a warthog-tick cycle is, therefore, probable [40,42]. Subsequent typing confirmed the presence of a genotype V virus in the warthog sampled from Tengani in southern Malawi in 1960 [47], genotype VIII in outbreaks in domestic pigs and associated ticks in 1978, 1984, 1986–1987, 1989–1992 [47,72] and genotype VIII in an *Ornithodoros* tick sampled in 1983 from a domestic pig pen [73]. Thus, whilst a historical domestic pig-tick cycle was evident, no studies documenting the presence of a warthog-tick cycle in Malawi have been produced.

Finally, the first historical record of apparent maintenance of ASFV circulation in a population of domestic pigs in Africa was an outbreak that was later confirmed as ASF in Angola in 1932 [74]. The source of the disease was believed to be local breed pigs kept in a traditional free-ranging husbandry system [74]. Subsequent virus typing of a 1970 outbreak strain confirmed the presence of genotype I [47]. Although warthog involvement has never been investigated in this country, it is likely that these pigs may have been exposed to ASF through the sylvatic cycle. Thus, the possibility that the warthog/tick sylvatic cycle still exists in Angola cannot be excluded, although information on this aspect is lacking.

### 2.2. Eastern Africa 

East Africa can be considered the real cradle of ASFV. Indeed, based on molecular clock studies, warthog populations from the Horn of Africa (*Phacochoerus aethiopicus*) have been maintaining ASF genes for more than 1.4 million years [75,76]. In addition to the seminal paper by Montgomery [3] in which the disease was first described from Kenya, the confirmation that warthog-burrow-associated *Ornithodoros* ticks were vectors of the disease was first reported in three East African countries (Kenya, Tanzania, and Uganda) in the late 1960s and early 70s [5,7,8,49]. These reports provided the impetus for subsequent studies that confirmed the presence of a sylvatic cycle in different wildlife areas in southern and East Africa. The presence of *Ornithodoros* ticks associated with warthogs was reported in Southern Kenya on several occasions [5,64] following initial attempts that failed to find ASFV in ticks from a warthog burrow [77]. Subsequent retrospective genetic studies have confirmed that genotype X is present in warthogs sampled in 1957 and 1959 [72], genotype I in bushpigs sampled in 1961 and genotypes X in tampans and IX in warthogs sampled in 2005 and again in 2008–2009 in Kenya (Table 1). Of interest is that domestic pigs sampled at the same time were also infected with genotype X, and all 51 warthogs were seropositive for ASFV.

In Uganda, Plowright et al. (1969b) reported 22% ASFV-positive warthogs in western Uganda (Queen Elizabeth National Park (NP), but no virus was obtained from ticks found in warthog burrows [5]. However, in a later study in Ruwenzori NP, 82% of warthogs were seropositive for antibodies to ASFV, with an extremely low infection rate of 0.017% in *Ornithodoros* ticks collected from warthog burrows [25]. Another study provided evidence of the occurrence of a sylvatic tick-warthog cycle in Lake Mburo NP [78]. Other authors have documented potential contacts between warthogs and domestic pigs at the edge of Murchison Falls NP and possible links with ASF outbreaks [79,80]. However, these studies are scattered in space and time, and there is no clear and updated information about the distribution of the sylvatic cycle in Uganda.

In Tanzania, the first reports of the occurrence of a natural sylvatic cycle in the Serengeti NP date from the late 1960s [68]. Subsequently, the continued presence of a very high proportion of serologically positive warthogs in the Serengeti NP was confirmed [25,33]. More recently, further investigations have detected the occurrence of a warthog-tick sylvatic cycle and the circulation of genotype XV in Saadani NP [32]. The presence of a sylvatic cycle has never been investigated in Burundi, Ethiopia, or Rwanda, but it is suspected to be present in all three countries, as all have reported outbreaks of ASF in domestic pigs.

### 2.3. West and Central Africa 

Ticks of the *Ornithodoros moubata* complex do not naturally occur in the area south of 13° N [81,82,83]. Therefore, it is believed that the natural warthog-tick sylvatic cycle is absent from the West and northern Central Africa region (Figure 3). This information was confirmed by a habitat suitability study, which suggested the distribution of ticks of the *Ornithodoros moubata* complex in an area circumscribed to East and Southern Africa [84]. Most of the Central African region encompassing the Congo Basin is represented by tropical forest habitat, which is not a suitable habitat for warthogs. However, warthog populations do occur in drier savannah areas of Central Africa, such as Southern DRC. Despite a 1978 record of infection in a warthog in the Republic of Congo [25], warthog populations are now believed to be extinct in that country [85]. A long history of ASF in domestic pigs and the presence of more than one genotype of ASFV in DRC are suggestive of the existence of a sylvatic cycle in this country [67].

In West Africa, despite the presence of warthogs in many natural savannah areas [85], a large-scale survey performed in an extensive part of the African continent a decade ago already documented that Sierra Leone, Ghana, and Nigeria had never reported the occurrence of ticks belonging to the *Ornithodoros moubata* complex [82]. In Nigeria, there was a single published report of the detection of ASF viral DNA in a baby warthog captured in the Adamawa region in 2008 [86]. Nevertheless, the absence of further reports to date makes it difficult to be confident that this observation reflects the reality in the field.

In drier Sahelian habitats of West Africa, such as Senegal, another species of argasid tick, *Ornithodoros sonrai*, colonizes rodent burrows in proximity to and within human habitations and is responsible for the occurrence of tick-borne human relapsing fever borreliosis. This species of *Ornithodoros* tick is widespread in West Africa, with 97% of the villages in some regions being infested [83,87]. The presence of this tick was investigated in 10 warthog burrows from the Siné Saloum protected area on the central western coast of Senegal and was found to be absent from those habitats, although its presence was confirmed in contiguous rodent or insectivore burrows. It was concluded that *O. sonrai* did not colonize warthog burrows and did not play a role in the occurrence of a potential sylvatic cycle of ASFV [88]. Subsequently, the occurrence of a sylvatic cycle was investigated through a serological survey for the detection of antibodies in six different warthog populations from Senegal and Southern Mauritania. The absence of ASF antibodies in this batch of 162 sera from five different locations in this region suggests the absence of a sylvatic cycle in this region [89]. Similarly, the absence of any reports of ASF in West and northern Central Africa until the 1950s, while the area was very rich in pigs, is another strong indicator of the absence of a sylvatic cycle in this region. Support for this hypothesis comes from a recent genetic study on warthog species which indicates that the warthog-tick cycle evolved in eastern Africa, initially within desert warthog (*P. aethiopicus*) populations, and that the common warthog (*P. africanus*) populations originating from West Africa, had not been exposed to ASFV until they interbred with East African desert warthog populations [76].

## 3. Major Gaps of Knowledge Regarding the Ecology and Distribution of the Warthog-Tick Sylvatic Cycle

### 3.1. Potential Drivers of the ASF Sylvatic Cycle Occurrence and Distribution

In areas with a well-established sylvatic cycle in East and Southern Africa, we can observe a series of predominant characteristic patterns. Those include high seroprevalence of antibodies against ASFV in warthog populations, which can range between 80 to 100% [58,90]. Such high seroprevalence is the result of exposure of warthog populations to the ASF virus through tick bites since infected warthogs do not excrete sufficient amounts of the virus for horizontal transmission between them to be efficient [6,52]. In contrast, the proportion of ASF-infected ticks collected in warthog burrows is quite low and usually below 5.1%, although this figure can be variable depending on the stage and the sex of the ticks and on the methods used to detect the virus [91]; (Table 1). Two recent studies report higher tick infection rates: The first one is a study in Saadani NP in Tanzania found that 18% of 111 adult ticks were positive for ASFV [32]. A second study found that 19% of the tick pools from warthog burrows and 15% in tick pools from pig pens in Gorongosa NP in Mozambique tested positive [29]. However, for the latter study, the adjusted infection rates, based on an assumption of one positive tick per tick pool, recovered a prevalence of 1.6% [91], which is consistent with the 0.06–5.1% range identified in all other studies pre-dating the Tanzanian study published in 2021 [24]. 

The number of burrows infested with ticks in those areas is considerable, and available data suggest it can range between 30% and 88% (see Table 1). Nevertheless, as research on the sylvatic cycle and associated methodologies have evolved, the number of exceptions to these trends is growing. In some cases, there are marked variations between areas that are geographically very close (Table 1), and in some cases, high ASF antibody prevalence in warthogs has been reported in areas where no soft ticks could be found in the burrows. In other cases, low seroprevalence levels have been observed in warthogs despite the presence of ticks in the burrows. Together this underscores previous reports that the presence of warthogs and ticks does not always coincide with the presence of virus in ticks [31].

Another aspect that is likely to influence the level of infestation of burrows with ticks and their level of ASF infection is the presence/absence of farrowing seasonality. In southern Africa, it is cyclical and usually occurs once a year, with births coinciding with the rainy season [92,93], whereas reproduction occurs year-round closer to the equator. During farrowing, females stay for several weeks in the burrows to take care of the young warthogs that become infected with the virus and develop a notable viremia. Therefore, it is likely that levels of ASFV amplify in resident ticks around this time [6].

The response of warthogs and their association with ticks to anthropogenic habitat transformation is largely unexplored but highly relevant in the context of climate change and natural habitat transformation [91]. *Ornithodoros* ticks are found in different kinds of warthog habitats. This includes burrows but equally other types of resting areas used by the species, including tree roots, rocky shaded areas (Jourdan, H., personal communication), and man-made structures. As an example, the use of concrete road culverts as resting places has been documented in several African countries, such as South Africa, Zambia and Tanzania [34], and Malawi [42]. This behavioral adaptation of warthogs to man-modified structures could potentially facilitate new spots for the proliferation of *Ornithodoros* ticks, suggesting that the natural sylvatic cycle is not solely reliant on the buffered burrow environment and may be resilient to future trends of environmental transformation in East and Southern Africa.

At the local scale, *Ornithodoros*’ presence and abundance are suspected to be influenced by altitude, pedology, hydrography, and vegetation, particularly surrounding warthog habitats [94]. A warthog can make use of multiple burrows, depending on its needs [85], and the number of burrows available per warthog is variable. In the Hluhluwe-Imfolozi NP in South Africa, with an estimated warthog density of 3.89 individuals/km^2^, the average number of burrows available per warthog ranged between one to four [92]. It is likely that the use of resting places or the availability of burrows is influenced by the type of soil and vegetation but also by the sympatric mammal community composition sharing some of those habitats [86], such as porcupines (*Hystrix cristata*) who are also secondary modifiers of burrows and which compete for burrow use, and aardvarks (*Orycteropus afer*), the primary excavator of burrows. Indeed, warthogs do not dig their own burrows but rather use those dug by these species. In addition, it is likely that porcupines and aardvarks act as suitable hosts for soft ticks in the burrows and, therefore, can play a role in tick population dynamics [94].

In many parts of Africa, warthog populations are threatened by human-caused habitat degradation, loss and fragmentation, and competition with livestock for water and food [85,95]. In this context of the proximity of warthogs to human habitats, the potential impact of environmental degradation in the natural sylvatic cycle has never been investigated. Environmental pollution of natural habitats with residues of pesticides or acaricides, for instance, could potentially affect soft tick survival. The extent to which some of these products could have an impact on *Ornithodoros’* survival or viability remains unexplored to date.

### 3.2. Tick Migration, Drivers and Its Impact on Population Genetics

*Ornithodoros* ticks are the main source of transmission of ASFV from warthogs to domestic pigs. However, the pathways of transmission between warthogs and domestic pigs are presumed to be due to tick transmission. This is based on experimental infection experiments, which demonstrated that direct contact transmission from viraemic warthogs to other warthogs or pigs did not occur despite sufficiently high levels of viraemia to infect ticks that fed on some of the infected individuals [52]. Feeding offal of acutely experimentally infected warthogs to domestic pigs resulted in an infection in some of the pigs when liquidized offal that included lymph nodes was used but was not considered likely to be important under natural conditions [52]. However, how this information translates into common practices or field situations is uncertain. For instance, the potential capacity of carcasses of infected individuals being scavenged by other wild or domestic pigs has never been explored by field experiments or computer modeling. Plowright et al. (1969b) reported that although ASFV was recovered from 22% of warthogs sampled in the Queen Elizabeth NP in Uganda, feeding lymphoid tissues to pigs failed to result in infection [5].

As *Ornithodoros* ticks are cryptic and photophobic, they rarely leave the burrows, feeding rapidly at night and then dropping off the warthog. In light of this, it was difficult to understand how transmission to domestic pigs in contact with warthogs was achieved. It was suggested that the ticks might occasionally be moved on warthogs or warthog carcasses, based on a report of *Ornithodoros* on warthogs shot early in the morning after exiting burrows. In addition, surveys of ectoparasites of warthogs in South Africa and Namibia confirmed that *Ornithodoros* nymphs travel on warthogs, sometimes in large numbers [60,61,62], providing an explanation for how they might be exposed to domestic pigs and indicating that there may be movement of ticks between burrows.

Genetic markers are needed to evaluate the importance of migration in *Ornithodoros* ticks and, consequently, the potential link via ticks between the sylvatic and the domestic cycles. Currently, there is no population genetics study on Afrotropical *Ornithodoros* species. However, studies performed on other tick species can provide information on what to expect in Afrotropical *Ornithodoros* ticks [96]. Based on the literature available for hard ticks, tick dispersal often depends on the type and mobility of the host, with high gene flow for tick species that feed on large and/or highly mobile hosts and low gene flow for tick species feeding on less mobile hosts. Afrotropical *Ornithodoros* are nidicolous and have different life history traits compared to hard ticks (cf. long lifespans, short feeding times), the way in which their populations move between burrows and the patterns of their gene flow in the natural environment or between the natural and the domestic pig compartments are largely unknown. Potentially, genetic markers can be designed to assess tick dispersal and population structure. Nuclear markers are preferred to mitochondrial ones, for which the resolution is often not sufficient to detect genetic variation at a population level [96].

## 4. The Sylvatic Cycle and Its Association with Virus Diversity

Molecular estimates of the integration of ASFV genetic material in the genomes of soft ticks of the *Ornithodoros moubata* complex suggest that this occurred at least 1.47 million years ago [75]. Such long-term co-evolution of soft ticks and warthogs with the virus has allowed the development of a high level of viral diversity that remains present today. Several studies in eastern and southern Africa have indicated that ASFV is maintained within sylvatic hosts, which have an essential role in the maintenance and evolution of the ASF virus [5,32,51,97,98]. In addition, molecular epidemiology using the *p72* gene has provided some indication of disease complexity on the African continent [47]. In addition, with the development of laboratories with higher genomic analytical performance, several African countries have been able to document more accurately the important viral diversity occurring in the warthog tick population infected with ASFV [22,23,36]. Table 3 summarizes known hosts and countries in Africa affected by the 24 known *p72* genotypes. 

Information collected to date suggests that virus diversity within the sylvatic cycle is high and is likely to be underestimated because this natural diversity remains unexplored for a large part of the continent. Similarly, we do not know the impact of fluctuations in tick dynamics on virus diversity and how these might affect the selection pressures that influence the survival and dissemination of certain virus strains within a pig environment and across the pig value chain.

## 5. The Domestic Pig-Tick Cycle

The domestic cycle between pigs and *Ornithodoros* ticks was first discovered in Spain after the introduction of ASF in 1960 [20]. It was found to be present in several districts in Spain and Portugal in the south-western part of the Iberian Peninsula and was responsible for prolonging the presence of ASF in that region [103]. An outbreak of ASF in the Alentejo region of Portugal in 1999, five years after the official eradication of the disease, was attributed to the presence of infected *Ornithodoros erraticus* ticks in a pig shelter into which pigs were introduced following the required period of depopulation [13]. 

In sub-Saharan Africa, this cycle has been very seldom investigated. Despite the fact that ASF in South Africa had been studied since 1928, the existence of a cycle between domestic pigs and ticks was not considered relevant at that time and was not investigated. However, Bedford (1936) reported the presence of large numbers of *Ornithodoros* ticks in pigsties in the ASF-endemic area in the north-eastern part of South Africa during a survey related to human relapsing fever [104], but the ticks were not examined for ASF virus. In 1985 a veterinarian reported finding large numbers of *Ornithodoros* ticks in a piggery in the same area in Limpopo Province. The ticks were not tested for ASFV, but the pigs on the farm were reported never to have experienced ASF symptoms [25]. 

After highlighting the potential role of soft ticks in the maintenance of ASFV in the Iberian Peninsula, the first and only in-depth investigation of the cycle of ASFV transmission between pig-adapted *Ornithodoros* ticks and ASF in Southern Africa was undertaken in an ASF-endemic area in the Mchinji district, located in Southwestern Malawi [27,40,41,42]. Such studies revealed ASFV-infected *Ornithodoros* ticks that lived in pig shelters [40]. Although the adjacent districts of Zambia and Mozambique are also endemic for ASF, to date, the domestic tick-pig cycle has not been confirmed to occur there [34,64]. Subsequently, ASFV-infected ticks were, however, reported in two pig pens from a village at the interface of Gorongosa NP in Central Mozambique [28,29]. 

One of the potential consequences of the presence of *Ornithodoros* ticks in pig premises is the regular exposure of pigs to the virus via tick bites, which could partially explain the increased resistance to the pathogenic effects of ASFV observed in some populations of pigs in several East and Southern African locations [21,42,64,98,105,106]. The link of this repeated exposure to ASFV with reports of low mortality ASF occurrence and cases of asymptomatic ASF cases in pigs is highly relevant and certainly deserves further investigation because it has only been studied on very limited occasions. This lack of knowledge affects not only the distribution of this pig-tick cycle but also the taxonomy and genetic characteristics of soft tick populations living in pig premises and their capacity to become infected and maintain ASFV populations and viral diversity when compared to the ones found in warthog burrows. 

The presence of *Ornithodoros moubata* complex ticks in Madagascar was investigated in association with foci of human relapsing fever in humans over several decades [107,108,109,110]. Further searches for soft ticks, including in pigsties, continued when outbreaks of ASF were confirmed for the first time and resulted in the finding of *Ornithodoros* ticks in a pigsty in Mahitsy in the Central Highlands northwest of Antananarivo [39]. The same location was revisited in 2007 and found to remain infested with soft ticks, 8% of which tested positive for ASF viral DNA, although the sty had remained empty since 2004 when the owner had stopped farming pigs due to repeated ASF outbreaks. Although only 5.6% of domestic pigs sampled tested positive for tick salivary antigens [38], a domestic cycle warrants further investigation, given the lengthy duration of ASF in Madagascar. 

*Ornithodoros* ticks are more likely to be found in basic pigsty constructions that offer shelter in crevices to ticks and, therefore, more commonly found in the backyard sector. Indeed, if pigsties do not offer any hiding places for the tick, i.e., they have solid floors and walls without any cracks and crevices, the ticks are unlikely to be present, and this is something that can and has changed over time [13]. Walton, in the early 1960s, already reported that improvements in human housing in Angola had resulted in the disappearance of *Ornithodoros* from human habitations, and, with it, the reduction of outbreaks of human relapsing fever [111]. It is likely that improvements to pig pens in Madagascar explain why no recent records of the ticks could be found in areas where they were historically present [38].

## 6. Current Knowledge and Gaps in *Ornithodoros* Tick Systematics, Biology and Life History Traits

### 6.1. Taxonomy and Diversity Ornithodoros spp. Populations

Since the first description in the literature of *O. moubata s.l.* in 1962 by Walton, the taxonomy of *Ornithodoros* species has been controversial due to the lack of adequate morphological criteria for identification and differing views on host specificity [111]. The first major step in addressing the controversies was made in 2009 using mitochondrial 16S rRNA sequencing, leading to confirmation of the monophyly of the *O. moubata* group and identifying three geographically discrete evolutionary lineages but showing that previous subspecies assignments based on host preference were not supported [51]. Using the same marker but a larger dataset, Bakkes et al. (2018) assigned novel species status to two of the evolutionary lineages described previously by Bastos et al. (2009) [112]. These newly-defined species, *O. waterbergensis* and *O. phacochoerus*, constitute a monophyletic group that includes *O. moubata*, *O. porcinus*, and *O. compactus* (a species associated with tortoises in South Africa). Subsequent molecular dating based on whole mitochondrial genomes and individual gene (18S rRNA and 28S rRNA) phylogenies indicate that the monophyletic lineage arose 4.2 ± 1 million years ago (MYA) [113]. 

The taxonomic diversity of *Ornithodoros* spp. found in warthog burrows in East and Southern Africa is shown in Figure 3; however, it must be noted that most *O. moubata s.l.* records were reported earlier than the extensive taxonomic revision of 2019 and cannot be assigned to specific species (Vial and Estrada-Pena, unpublished data). Future studies on warthog burrow-associated soft ticks require molecular species identification since discriminant morphological characters are scarce and not straightforward. In addition, the biological validity of the newly-proposed species requires confirmation through breeding experiments.

### 6.2. Pathogenesis of Ornithodoros Ticks and Vector Competence

In principle, any blood-sucking insect biting an infected animal can potentially transmit the virus if it subsequently bites a susceptible pig. However, in contrast to long-lived *Ornithodoros* ticks, most biting insects are not believed to play a major role in virus transmission. *Ornithodoros* ticks are considered a valid biological vector for ASF because they can maintain the virus after an initially infected suid blood meal and subsequently transmit it vertically and horizontally to its progeny and to a new host. In addition, because they are long-lived, multiple feeding and virus transmission events to warthogs are possible. However, as alluded to previously, there is some variation in virus maintenance efficiency between different *Ornithodoros* tick species. More precisely, the demonstration of vector competence requires following several laboratory steps with captive vector collections under controlled experimental conditions. Those include confirming the following aspects: the ability of the vector to ingest infected blood, longitudinal persistence of the virus through time, and dissemination of the pathogen to vector organs involved in horizontal (salivary glands) or vertical transmission (reproductive organs). These methods have enabled confirmation that *Ornithodoros* species are biological vectors and reservoirs for some ASFV strains in southern and eastern Africa and the Iberian Peninsula [5,6,14,16,33].

ASFV replication and dissemination in ticks are critical parameters to understand vector competence but not the only ones. Some studies demonstrated that tick competence for ASFV can vary according to the ASFV strain. For example, *O. coriaceus* was able to transmit to pigs the Tengani/62 but not Uganda/61 strain, while *O. porcinus* transmitted both strains [15,114]. The same observation was published for vertical transmission for O. moubata transmitting the Liv13/33 but not the Georgia 2007/1 to the descendants [18]. 

Despite the fact that soft ticks can act as reservoirs of ASFV, the virus can be pathogenic to tick populations, as suggested by observations of high mortality among O. moubata populations infected with some strains of genotype I (VIC T90/1 and Liv13/33) or genotype II (Georgia2007/1) [18,115]. The virus was always detected in dead ticks despite their relatively advanced state of decomposition [116]. In addition, the same strains could induce different levels of mortality rates depending on the tick species they infected [19]. Studies of the tick microbiome suggest that interactions with other pathogens might affect competence as vectors of ASFV and require further investigation. These kinds of investigations implemented for other tick vectors [117,118] have yet to be explored in *Ornithodoros* tick populations.

## 7. Improved Methods and Approaches for Assessing the Occurrence of *Ornithodoros* Ticks as Reservoirs of ASFV

Several countries in Africa have to date, never confirmed the presence or absence of a sylvatic or domestic cycle involving ticks in their territory. This should be investigated, especially in countries where these cycles are likely to occur, using some of the methods that are described below.

### 7.1. Direct Methods

Direct methods are those that enable verification of the presence of ticks in warthog burrows from the African natural habitats or pig sties. Those include manual collection and carbon dioxide (CO_2_) traps. 

#### 7.1.1. Manual Collection

Monitoring ASF infection in *Ornithodoros* ticks from warthog burrows has proven to be a useful and inexpensive alternative to monitoring evidence of infection in warthogs [89]. It is the most widespread method to investigate the presence of ticks in a given environment and involves removing the soil from the bottom and walls of specific habitats with a shovel. The soil, if sandy, may be sieved over a white tray or plastic sheet using a stainless-steel Labotec test sieve with apertures of 3350 µm. The soil is then left exposed to the sunlight to facilitate the detection of small ticks that pass through the sieve or those large ticks that are retained by the sieve. If ticks are present, they are detected when they start moving after a few seconds of sunlight exposure. The method is easy to implement and does not require sophisticated material. However, in order to make results comparable in different study sites, the following guidelines are recommended: 

Details of the effort implemented to detect warthog burrows, such as time of the search, means of transport for burrow search (walking, ground vehicle, flying, etc.), and the number of persons involved should be recorded. Similarly, the same tick search effort should be implemented on every burrow so that the number of ticks collected is comparable between burrows or pigsties. This effort time should include soil collection but also the time of observation for tick detection and collection.

The manual collection is more easily applied to targeted locations with a reduced searching surface (warthog burrows) than to more extensive surfaces occupied by domestic pigs where it might be difficult to decide on specific search order or direction.

#### 7.1.2. Carbon Dioxide Traps

Carbon dioxide traps are known to be a useful method to attract and capture ticks and other arthropods from animal burrows [119]. Caiado et al. (1990) tested this method successfully to collect *Ornithodoros erraticus* ticks in the Portuguese traditional pigsties, which proved to be both simple and more effective than manual collection [120]. In Africa, Nevill tested CO_2_ traps successfully on *Ornithodoros savignyi* in the Kalahari [121]. Despite this method showing a lot of potential, no published studies are available to date describing *Ornithodoros* tick collection from warthog burrows, pigsties or other potential ASF host habitats in Africa or Madagascar. However, some investigations are ongoing to evaluate and standardize this tick-collection approach. The main limitation of its use is the availability and conservation of dry ice used to produce carbon dioxide in remote areas.

### 7.2. Indirect Methods

Indirect methods can provide a proxy for the presence of soft ticks in a given wild or domestic environment. Those include the detection of antibodies against the virus in warthogs or against salivary antigens of ticks in domestic pigs [89].

#### 7.2.1. Detection of ASFV Antibodies in Warthogs and Pigs

Although warthogs do not develop sufficient viremia to shed ASFV [52], they develop long-life antibodies that can be easily detected by standard commercial ELISA kits, and when the sylvatic cycle is present in the natural environment, it is common that a large proportion of the warthog population develops antibodies against ASF virus. As seen in Table 3, it is common that this proportion is above 70% of the sampled population. The main limiting factor to undertaking such an activity used to be the availability of samples that required capturing the animals. However, since warthogs are a popular game species, it is possible to benefit from hunting activities to obtain hunting samples. The possibility of using filter papers to monitor ASF antibodies by ELISA or even detect ASFV DNA by PCR [122] has rendered this activity easier to implement than in previous years since the collection of blood with filter paper does not require any specific expertise and the samples can be stored at room temperature [122,123].

#### 7.2.2. Detection of Antibodies against *Ornithodoros* Salivary Components

Vertebrate hosts are able to develop an immune response when they are bitten by a tick, which can be used as a marker of tick exposure. Those antibodies can be detected by immunological methods such as ELISA tests. The first anti-tick ELISA test was developed in Spain at the end of the 20th century [124] and has been successively improved to increase its specificity [125,126]. Field tests have been implemented in areas where the sylvatic cycle was present with positive results [28]. However, residual percentages of positivity in areas where *Ornithodoros* ticks are scarce [102] or unlikely to occur [44,127] suggest that cross-reactions are still occurring, and false positive results can be misleading. When a low percentage of samples are positive, it is necessary to confirm the presence of the ticks before assuming tick involvement in ASF.

#### 7.2.3. Historical Records of Previous ASF Outbreaks in Domestic Pigs

If pig farming is practiced in basic facilities and ticks are present, it is likely that local populations are confronted with the occurrence of regular outbreaks of ASF. In those cases, forms of the disease tend to be milder, with the common occurrence of low mortality outbreaks and observations of asymptomatic cases, which lead to higher chances of detecting seropositive pigs. In those cases, the detection of antibodies against ASF by serological methods combined with a history of regular outbreaks can be used as an additional indicator of the potential presence of *Ornithodoros* ticks living in the environment under study.

This panel of direct and indirect methods can provide information on the presence or absence of ASF cycles involving ticks. Given the diversity of options, it is recommended to use them in combination in order to maximize the number of indicators that can help to confirm the occurrence of ASF vectors in a given environment [30].

## 8. Conclusions

Despite the global importance reached by ASF in the last decades, the role of soft tick in the epidemiology and ecology of this virus in Sub-Saharan Africa still has many knowledge gaps and offers many opportunities for further investigations. This is illustrated by the amount of newly discovered ASFV genotypes that seem to be circulating within the sylvatic cycle, and that continues to grow with every new investigation. Addressing this lack of knowledge is even more urgent in the context of a booming African pig population to satisfy the needs of exponential human demographic growth in the continent. Current trends are likely to facilitate increasing contact between domestic pigs and *Ornithodoros* ticks in many regions of East and Southern Africa. The potential impact of such contacts on the genetic variability and evolution of ASF virus strains in the domestic pig value chain and their capacity to generate mild or sub-clinical forms of the disease in local domestic pigs remains unknown and clearly justifies further research efforts across the continent.

The current review has clearly highlighted that the available knowledge on the distribution of *Ornithodoros* tick populations across the African continent is very patchy and irregular. Indeed, the presence of a natural sylvatic cycle in many countries and regions has never been explored; in others, knowledge of the distribution is very limited. In addition, for some areas, its presence has been documented in the past, but recent assessments are lacking, and much needed given changing climatic conditions and human practices. In this context, understanding the impact of such drivers is crucial, and periodic reassessment becomes necessary. Indeed, the environmental drivers and anthropogenic changes that can affect tick abundance and their influence on ASF virus presence and diversity have received very little attention and are largely unknown. This lack of knowledge is even more serious in the case of the domestic pig-tick cycle, which has only been documented on very rare occasions, and, sometimes, decades ago. Gathering this information would allow us to identify hot spots of virus diversity and the emergence of new strains and equally hyper-endemic areas of ASF with a high prevalence of mild forms of the disease in order to target monitoring and control efforts. However, despite global warming and climate change, there is no current evidence that the distribution of *O. moubata* complex ticks is expanding to new territories beyond East and Southern Africa.

In summary, recent investigations on the entomological aspects of ASF have highlighted the fact that our understanding of the role of soft tick populations in the maintenance dissemination and evolution of ASF remains limited, and a lot of research gaps remain to be explored. Considering the current trends of population growth, agricultural expansion, habitat transformation, climate change, and ASF expansion, these questions deserve higher consideration in the next years, if we want to be more efficient in the control of this disease

## Figures and Tables

**Figure 1 pathogens-12-00469-f001:**
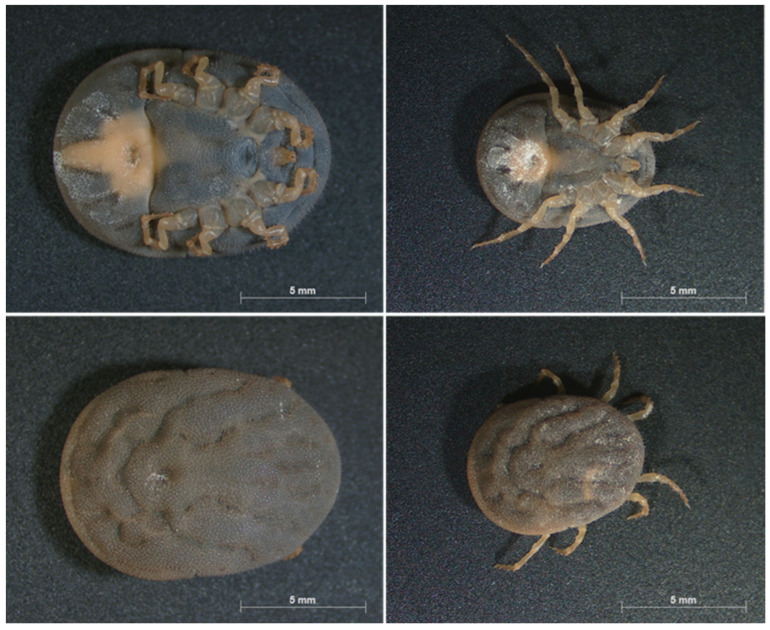
Photos of dorsal and ventral view of female (**left**) and male (**right**) *Ornithodoros moubata* (Source: F. Taraveau). Ticks are from the Neuchâtel strain (University of Neuchâtel, Switzerland), maintained at the insectarium from CIRAD (Montpellier, France) since 2008.

**Figure 2 pathogens-12-00469-f002:**
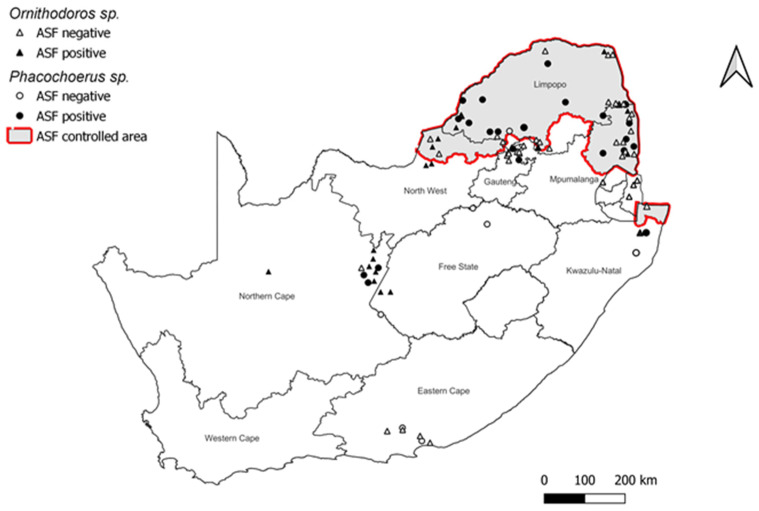
Map of South Africa representing the ASF controlled area and the locations where infected warthog and soft tick populations have been investigated and detected (Source: K. Rea).

**Figure 3 pathogens-12-00469-f003:**
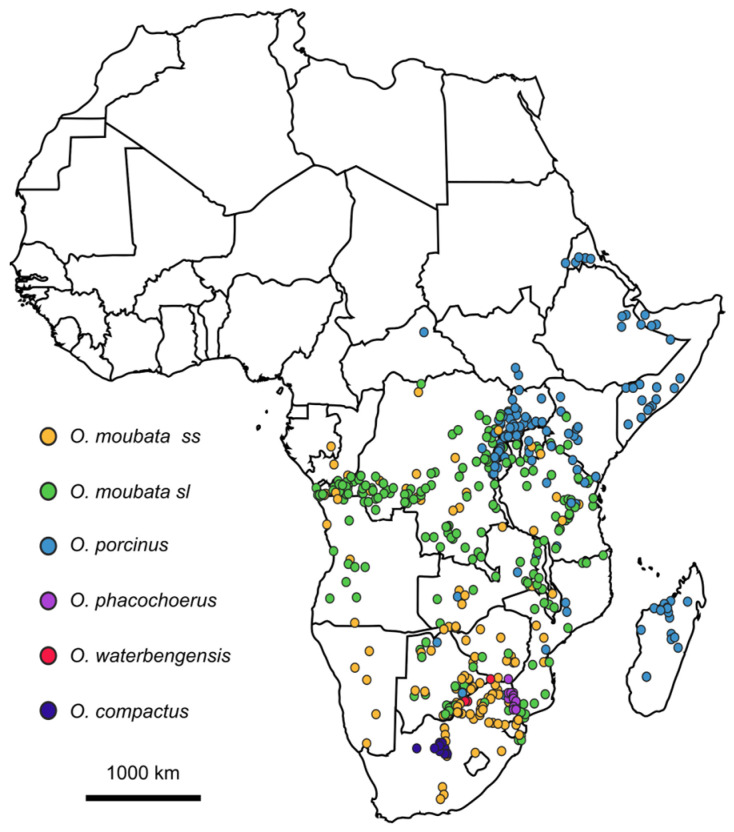
Geographical localization of presence data for the Afrotropical soft tick vector of ASFV based on 574 occurrences published between 1873 and 2022 in East and Southern Africa. Species identification appears as referenced by the different authors (Source: H. Jourdan-Pineau).

**Table 1 pathogens-12-00469-t001:** Summary of the available information regarding the level of seroprevalence of warthog populations, the level of infestation of warthog burrows, and the level of ASF virus found in soft ticks collected from burrows in different study sites in East and Southern Africa.

Tab	Location	Warthog ASF Seroprevalence (% Positive)	Percentage of Infested Burrows (%)	Percentage of Burrows with ASFV-Infected Ticks	Percentage of Positive Ticks (%)	Reference
Kenya	Mara, Nairobi, Nguruman	100% (n = 8)	30% (n = 147)	NA	0.440% (n = 10,393)	[25]
	Lolldaiga	75% (n = 4)	0	NA	0	[25]
	Nguruman	NA	95% (n = 20)	30% (n = 20)	0.200% (n = 7509)	[5]
	Machakos	100%	NA	NA	22% * (n = 1576)	[26]
Malawi	Liwonde National Park	NA	100 (n = 1)	0% (n = 1)	0% (n = 1400)	[27]
Mozambique	Gorongosa NP	75% (n = 12)	90% (n = 31)	72.4% (n = 31)	Not investigated	[28,29]
South Africa	Kruger NP	93% (n = 73)	55% (n = 18)	NA	1.390% (n = 1026)	[6]
	North Transvaal (Lat < 25°)	92% (n = 51)	44% (n = 36)	NA	0.321% (n = 14,023)	[25]
	North Transvaal (Lat < 25°)	4%	0	NA	1.740% (n = 980)	[25]
	Mkuze Game Reserve	4% (n = 260)	33% (n = 40)	NA	0.060% (n = 5018)	[25]
	Hluhluwe-Umfolozi	0% (n = 297)	0 (n = NA)	NA	0	[25]
	Ndumu Game Reserve	NA	0% (n = 35)	NA	NA	[30]
	Mkuze Game Reserve	NA	60% (n = 98)	0%	0%	[31]
Tanzania	Saadani NP	84% (n = 19)	41% (n = 12)	41% (n = 12)	18% (n = 111)	[32]
	Serengeti	100% (n = 108)	88% (n = 112)	NA	0.450% (n = 50,043)	[25]
	Serengeti	71%	85% (n = 40)	47% (n = 34)	0.650% (n = 8348)	[33]
Uganda	Ruwenzori	82%	65% (n = 71)	NA	0.017% (n = 64,599)	[25]
	Queen Elizabeth NP	22 (n = 103)	100% (n = 8)	0 (n = 8)	0 (n = 15,505)	[5]
Zambia	Kafue NP	NA	58.5% (n = 70)	35% (n = 20)	1.300% (n = 1476)	[34]
	Livingstone NP	NA	16% (n = 18)	100% (n = 18)	5.100% (n = 1133)	[34]
	South Luangwa	NA	36% (n = 28)	80% (n = 10)	1.400% (n = 588)	[34]
	Sumbu NP	NA	62% (n = 35)	30% (n = 10)	2.100% (n = 481)	[34]

Key: NA: Unreported or not investigated; n: sample size). * Presence of a warthog-tick cycle is assumed if Ornithodoros ticks from warthog burrows tested positive for ASFV.

**Table 2 pathogens-12-00469-t002:** Summary of first or most recent published information on *Ornithodoros*-related ASF in the Southern African and Indian Ocean region (Comoros and Seychelles are excluded as ASF has never been reported, no warthogs are present, and in Comoros, there are no pigs).

	Warthog-Tick Cycle *	Pig-Tick Cycle	References
Angola	Not investigated	Not investigated	No published information
Botswana	Serological evidence	Not investigated	Jori, 2015, unpublished data [35]
DRC	Not investigated	Not investigated	No published information
Eswatini	Probably absent	Not investigated	[36]
Lesotho	Not investigated	Not investigated	No published information (no ASF)
Madagascar	Absent (no warthogs)	Currently rare	[37,38,39]
Malawi	Historically present	Historically present	[27,40,41,42,43]
Mauritius	Absent (no warthogs)	Absent (no ticks)	[44,45]
Mozambique	Present	Present	[28,29]
Namibia	Present	Not investigated	[6,46,47,48]
South Africa	Present	Not investigated	[6,46]
Tanzania	Present	Not investigated	[7,8,32,33,49]
Zambia	Present	Not demonstrated	[23,34,50]
Zimbabwe	Present	Not investigated	[6,46,51]

Key: * Presence of a warthog-tick cycle is assumed if *Ornithodoros* ticks from warthog burrows tested positive for ASFV.

**Table 3 pathogens-12-00469-t003:** Country, *p72* genotype, and hosts of African swine fever virus identified in *Ornithodoros* ticks in Africa and Madagascar. The ‘other hosts’ indicated refer to the hosts associated with the genotype and not the specific country.

Country	Genotype	WH-T	DP-T	References	Other Hosts	Reported Distribution in Africa
Kenya	X	+	−	[26]	WH, DP	East Africa
Madagascar	II	−	+	[99]	WH, DP	East and southern Africa, Nigeria
Malawi	VIII	−	+	[72,73,100]	DP	East and southern Africa
Mozambique	II	+	−	[29]	WH, DP	East and southern Africa, Nigeria
Mozambique	V	+	−	[29]	WH, DP	Malawi, Mozambique
Mozambique	XXIV	+	+	[29]	−	Mozambique
South Africa	I	+	−	[22,59]	WH, BP, DP	Sub−Saharan Africa
South Africa	III	+	−	[22,59,101]	DP	Botswana, South Africa
South Africa	IV	+	−	[22,47]	DP	South Africa
South Africa	VIII	+	−	[22]	DP, DP-T	East and southern Africa
South Africa	XIX	+	−	[22]	DP	South Africa
South Africa	XX	+	−	[22,59,102]	WH, DP, EWB	DRC, South Africa
South Africa	XXI	+	−	[22,59]	WH, DP	South Africa
South Africa	XXII	+	−	[22,59,101]	WH, DP	South Africa
South Africa	XXV	+	−	Unpublished data	WH	South Africa
Tanzania	X	+	−	[72]	WH, DP	East Africa
Tanzania	XV	+	−	[32]	DP	Tanzania
Zambia	I	+	−	[101]	WH, BP, DP	Sub-Saharan Africa
Zambia	II	+	−	[72]	WH, DP	East and southern Africa, Nigeria
Zambia	VIII	+	−	[47]	DP, DP-T	East and southern Africa
Zambia	XI	+	−	[72]	−	Zambia
Zambia	XII	+	−	[72]	DP	Malawi, Zambia
Zambia	XIII	+	−	[72]	−	Zambia
Zambia	XIV	+	−	[72]	DP	DRC, Zambia
Zimbabwe	I	+	−	[47]	WH, BP, DP	Sub-Saharan Africa

Key: WH-T = warthog-tick; DP-T = domestic pig tick; WH = warthog; DP = domestic pig; BP = bushpig; EWB = Eurasian wild boar; DRC = The Democratic Republic of Congo.

## Data Availability

Not applicable as this is a review.

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
