# Peer review of "An Updated Review of Ornithodoros Ticks as Reservoirs of African Swine Fever in Sub-Saharan Africa and Madagascar"

_pathogens, 2023, doi:10.3390/pathogens12030469_

Round 1

Reviewer 1 Report

Line 89: suggest rewording to "These laboratory studies confirmed O. moubata longevity (insert # of years here), which has been.."

Line 90 reform into separate sentences: Although most of the Ornithodoros species studied experimentally have been able to maintain ASFV and transmit it to pigs; transovarial transmission has not been demonstrated in all species evaluated to date [14-92 16]. Vector competence remains a topic for investigation [17-19]and is discussed in 93 more detail in section 5.2.

Line 95 In the past few decades, the domestic pig population on the African continent has grown exponentially,

Table 1 seems to have lost some call darkening in the bottom left corner

Line 134 change to Southern Africa

Line 251: reword. Similarly, Lesotho, is situated within the borders of South Africa in an area far south of the ASF controlled area and is mostly bordered by mountains. Therefore, the likelihood of finding a sylvatic cycle in this country is very small.

Line 322: Ticks of the Ornithodoros moubata complex do not naturally occur in the area south 322 of 13°N [81-83]. You do not mention this complex up to this point. I think you should define what this is earlier in the manuscript. 

Line 469. Now you italicize Ornithodoros. Go back through the manuscript and italicize all the scientific names

Line 718 This section is Despite the global importance reached by ASF in the last decades, the 718 role of soft ticks in the epidemiology and ecology of this virus in Sub-Saharan Africa still 719 has many knowledge gaps and offers many opportunities for further investigations.

Author Response

My response to the comments of reviewer 1 is in the attached file

Reviewer 2 Report

In the article entitled: " An updated review of Ornithodoros ticks as reservoirs of African swine fever in Sub-Saharan Africa and Madagascar. " authors provided an interesting review gathering knowledge of the aspects regarding ASF vector – ticks from Ornithodoros spp.

African swine fever became a serious treat for pig industry worldwide. There are many knowledge gaps in the transmission of ASF – among them role of the possible vectors. Since role of the hard ticks was neglected before by several authors, the virus may replicate in soft ticks and play a role in distribution of the pathogen.

Introduction is sufficient, the review is valuable and comprehensive, however some minor remarks should be included.

Please find below minor comments:

1.       Line 5: please correct the name and M.L.

2.       Lines: 43-46, 46-50, 50-53: missing reference

3.       Please check the Table 1: % suffix is missing in several rows

4.       Line 130: reveal the abbreviation SADC

5.       Despite the title of the review, authors should consider the discussion about Ornithodoros tick distribution in Europe regarding the climate changes – which might be a good opportunity to consider this treat for EU countries.

6.       Line 365-370: Did authors considered that high seroprevalence in warthogs may be the result of exclusive transmission between the warthogs and not necessary between tick and warthog. This why the prevalence in ticks and warthogs may differ. Please discuss.

7.       Line 429-430: not clear, please reveal.

8.       Line 680-682: I would not agree. Low level of the viremia may be caused by the specific ASFV strain itself – circulating attenuated strains may produce a very low level of viremia. There are some studies indicating that low levels of virus may be found in non-invasive samples like saliva or faeces but also as the result of infection of attenuated strains. That’s why serologic tests are more adequate to investigate them. Although, the role of soft ticks cannot be neglected, since ASF is present in Africa for long years and evolution of ASFV strains into attenuated is possible.

9.       Line 718: delete “This section is”

I’ve got no further comments.

Author Response

Response to the reviewers can be found in the uploaded file.
